# CHANG-ES: XVIII—The CHANG-ES Survey and Selected Results

**Judith Irwin** [1,*], **Ancor Damas-Segovia** [2], **Marita Krause** [3], **Arpad Miskolczi** [4], **Jiangtao Li** [5], **Yelena Stein** [4,6], **Jayanne English** [7], **Richard Henriksen** [1], **Rainer Beck** [3], **Theresa Wiegert** [1] **and Ralf-Jürgen Dettmar** [4]

1   Department of Physics, Engineering Physics & Astronomy, Queen's University, Kingston, ON K7L 3N6, Canada; henriksn@astro.queensu.ca (R.H.); theresa.wiegert@gmail.com (T.W.)
2   Departamento de Astronomía Extragaláctica, Instituto de Astrofísica de Andalucía, Glorieta de la Astronomía sn, 18008 Granada, Spain; adamas@iaa.es
3   Max-Planck-Institut für Radioastronomie, Auf dem Hügel 69, 53121 Bonn, Germany; mkrause@mpifr-bonn.mpg.de (M.K.); rbeck@mpifr-bonn.mpg.de (R.B.)
4   Faculty of Physics & Astronomy, Astronomical Institute, Ruhr-University Bochum, 44780 Bochum, Germany; miskolczi@astro.rub.de (A.M.); ystein@astro.ruhr-uni-bochum.de (Y.S.); dettmar@astro.rub.de (R.-J.D.)
5   Department of Astronomy, University of Michigan, 311 West Hall, 1085 S. University Ave., Ann Arbor, MI 48109, USA; jiangtal@umich.edu
6   Observatoire Astronomique de Strasbourg, Université de Strasbourg, CNRS, UMR 7550, 11 rue de l'Université, F-67000 Strasbourg, France
7   Deptartment of Physics & Astronomy, University of Manitoba, Winnipeg, MB R3T 2N2, Canada; jayanne_english@umanitoba.ca
*   Correspondence: irwinja@queensu.ca

**Abstract:** The CHANG-ES (Continuum Halos in Nearby Galaxies) survey of 35 nearby edge-on galaxies is revealing new and sometimes unexpected and startling results in their radio continuum emission. The observations were in wide bandwidths centred at 1.6 and 6.0 GHz. Unique to this survey is full polarization data showing magnetic field structures in unprecedented detail, resolution and sensitivity for such a large sample. A wide range of new results are reported here, some never before seen in any galaxy. We see circular polarization and variability in active galactic nuclei (AGNs), in-disk discrete features, disk-halo structures sometimes only seen in polarization, and broad-scale halos with reversing magnetic fields, among others. This paper summarizes some of the CHANG-ES results seen thus far.

**Keywords:** galaxies individual; galaxies spiral; galaxies magnetic fields; radio continuum galaxies

## 1. Introduction to the CHANG-ES Survey

### 1.1. Motivation

CHANG-ES (Continuum Halos in Nearby Galaxies—an EVLA Survey) is a project to observe 35 edge-on galaxies in the radio continuum with wide bandwidths centred at 1.6 GHz (L-band) and 6.0 GHz (C-band, using the Karl J. Jansky Very Large Array (hereafter, the VLA, formerly the Expanded Very Large Array or EVLA). The wide bandwidths, $\Delta \nu$, of the VLA (Table 1) have made such a survey feasible since the theoretical signal-to-noise (S/N) improves as $1/\sqrt{\Delta \nu}$, all else being equal. For CHANG-ES, this has meant an order of magnitude improvement, on average, compared to any similar survey. For example, [1] achieved sensitivities ranging from 80 to 130 µJy beam$^{-1}$ rms and [2] achieved 45 to 90 µJy beam$^{-1}$ rms at 5 GHz. By contrast, typical rms values of the

CHANG-ES D-configuration images are $\approx 6\ \mu$Jy beam$^{-1}$ at the same frequency (range of values for all configurations and frequencies are summarized in Table 1). Additionally, no previous survey has included polarization. This has opened up the possibility of exploring the magnetic fields in galaxy disks and halos for a well-defined galaxy sample.

Wide bandwidths, when broken up into many individual channels, also provide advantages other than improved S/N: (a) radio frequency interference (RFI) can be more readily identified and excised, (b) *in-band* spectral index maps can be formed (e.g., Section 2.1.1), (c) Rotation Measure (RM) analysis can be carried out (Sections 2.4 and 2.5), and (d) imaging can make use of the power of multi-frequency synthesis.

**Table 1.** Overview of galaxy and observing parameters over frequencies and VLA array configurations.

| Parameter | Value | Ref |
|---|---|---|
| Galaxy Distance range | $4.4 \rightarrow 42$ Mpc | [3] |
| Galaxy SFR range | $0.02 \rightarrow 7.29\ M_\odot/$yr | [3] |
| Galaxy optical diameter range | $3.9 \rightarrow 15.8$ arcmin | [4] |
| Array configurations (L-band, C-band) | B C D, C D | [3] |
| $\nu_0$ (L-band, C-band) | 1.58, 6.00 GHz | [3] |
| $\Delta\nu$ (L-band, C-band) | 512 MHz, 2.0 GHz | [3] |
| No. of spectral channels (L-band, C-band) | 2048, 1024 | [3] |
| Approx. Spatial resolution range | $3 \rightarrow 60$ arcsec | [3,5,6] |
| Typical resolutions in arcsec (high-res, low-res uv weightings): | | |
|     B configuration L-band | 3, 6 | |
|     C configuration C-band | 3, 6 | |
|     C configuration L-band | 10, 15 | |
|     D configuration C-band | 10, 15 | |
|     D configuration L-band | 35, 45 | |
| Approx. rms range (all configurations and bands) | $3 \rightarrow 95\ \mu$Jy/beam | [3,5,6] |

CHANG-ES has a variety of goals. We wanted to understand the incidence of radio halos in 'normal' spiral galaxies, determine their scale heights and correlate these results with other properties such as the underlying star formation rate (SFR) or SFR 'surface density' (SFR per unit area, SFR$_{SA}$). Measurements of outflow speeds can help to determine whether galaxies are more likely to experience winds or 'fountain' flow. Outflows in nearby galaxies could also represent low-energy analogues of the 'feedback' that appears to be required in galaxy formation scenarios in order to control the star formation rate [7]. Since outflows can be observed with spectacular resolution in nearby galaxies, the physics of such systems may provide meaningful constraints on galaxy formation in the early universe. A consistent galaxy sample, with complementary data at other wavelengths, can also help us explore pressure balance in the ISM. For more information on our science goals, see [4].

As indicated above, unique to the survey is that all polarization products were measured, namely Stokes I, Q, U, and V. With analysis of linear polarization ($P = \sqrt{Q^2 + U^2}$), along with RMs, an in-depth understanding of the magnetic fields in these galaxies can be obtained, and several examples (Section 2.5) have been very illuminating in this regard. Some of our galaxies show lagging halos (e.g., see others [8]) either in optical emission lines or HI. The reason for such lags is not yet understood, but an association with magnetic fields is possible [9].

A survey size that samples a sufficient number of galaxies in a reasonable (but large) total observing time resulted in 35 galaxies in total. Clearly, rms values that improve upon previous measurements were desired. Since vertical features in galaxies show structures on virtually all scales, we required observations over a range of VLA array configurations. Thus, we have observed at B, C, and D-configurations in L-band and in C and D-configurations in C-band. This also provides us with matching resolution for spectral index maps which are essential in understanding the origin of the emission (e.g., thermal or non-thermal fractions) as well as the propagation of cosmic ray electrons (CREs).

Our galaxy selection criteria were (1) an inclination $> 75°$ as given in the Nearby Galaxies Catalog [10] in order to easily see a halo, (2) a declination $> −23°$ for detectability at the VLA, (3) an optical size between 4 and 15 arcmin for a good match to the array configurations, and (4) flux densities at L-band of at least 23 mJy in order for a detection to be more likely. Two more galaxies were added that did not meet these criteria but were included because of other interesting known halo properties (NGC 5775 which was smaller than the lower angular size limit, and NGC 4244 which had a flux density below the minimum cut-off).

Notice that selection effects should be minor, the main one being the possibility that, by adopting a radio flux density limit, we could inadvertently be choosing galaxies with high SFRs. However, Table 1 reveals that our SFR range is quite typical of 'normal' spirals. We do *not* choose our galaxies based on whether or not they have halos (except for the two mentioned above) nor do we specifically choose any active galactic nuclei (AGNs) although several were found to be present once the sample was defined. Table 1 provides a summary of our data. In all, over 405 hours of VLA time were granted for this large project.

*1.2. Techniques and Data Products*

Data reduction was carried out using the Common Astronomy Software Applications (CASA) package[1]. This allowed us to take advantage of a number of sophisticated imaging algorithms such as multi-scale multi-frequency synthesis (ms-mfs, [11]). 'Multi-scale' assumes that any intensity feature can be represented by spatial scales of varying width prior to convolution with the synthesized beam, as opposed to the traditional assumption that an intensity feature is represented as a series of points. 'Multi-frequency' employs the technique of sampling differing spatial scales as the frequency varies across the band, thereby more thoroughly filling in the uv plane. In addition, these techniques, as employed in CASA, allow for the fitting of in-band spectral indices across any individual band. In principle, one can solve for curvature, $\beta$, in the spectral index as well (i.e., $I_\nu \propto \nu^{\alpha + \beta log\nu}$ but our tests showed that the S/N for any given data set was not high enough to take advantage of the curvature option [12]. Hence, a straight line in log space ($\beta = 0$) was fitted across the band while imaging. Thus, a solution for the in-band spectral index, $\alpha$, was obtained point-by-point, in each galaxy.

At the time of writing, D-configuration images [3] have been released at our data release website[2], the B-configuration images have also been released [5] and the C-configuration [6] data release is imminent. Release data include total intensity images at two different uv weightings, i.e., 'robust 0' and 'robust 0 + uv taper', specified as high-res and low-res, respectively, in Table 1. Both non-primary-beam-corrected and primary-beam-corrected versions are included. At D-configuration, linear polarization and polarization angle maps, and in-band spectral index maps are also available. At high resolution, where emission is weaker, we opted to include band-to-band spectral index maps, rather than in-band spectral index maps, i.e., C-configuration C-band to B-configuration L-band $\alpha$ maps.

Because the largest angular size detectable at the VLA is 16 arcmin at L-band and 4 arcmin at C-band, it was necessary to supplement the VLA data with single-dish data for the largest galaxies at L-band and all galaxies at C-band (Table 1). This has been accomplished with over 200 h of supplementary Greenbank Telescope (GBT) observations. Because the GBT wide-band instrumentation was delayed and because data reduction algorithms had to be developed, all GBT observations are completed but the data reduction is currently in progress. For several galaxies (NGC 891, NGC 4565, and NGC 4631), Effelsberg data have been used for the large-scale total intensity emission.

Complementary to our radio data, we have obtained H$\alpha$ images using the Apache Point Observatory 3.5-m telescope [13] for 21 galaxies, supplemented with images from the literature for

---

[1]    http://casa.nrao.edu
[2]    https://www.queensu.ca/changes

the remaining galaxies. This data set was obtained to assist with the thermal/non-thermal separation analysis (Section 2.2) and will also soon be publicly released. We also have WISE enhanced resolution images of all galaxies and XMM-Newton and Chandra data for a large fraction of the sample.

## 2. CHANG-ES Highlights

In the following sections, we review CHANG-ES results. Rather than presenting the various papers in chronological order, we present the results topically.

### 2.1. AGNs

As indicated in the introduction, we did not select AGNs in the CHANG-ES sample nor was the sample designed specifically for AGN detection. However, galaxies with AGNs have turned out to provide some of the more interesting results to date, as follows.

#### 2.1.1. A Remarkable Nearby TDE: NGC 4845

Observations of NGC 4845 resulted in a serendipitous radio detection [14] of a Tidal Disruption event (TDE) approximately one year after its hard X-ray outburst [15]. These authors attribute the X-ray variability to a super-Jupiter mass object being devoured by a $10^5$ $M_\odot$ black hole. CHANG-ES detected the radio emission as well as the *variability* of the radio emission from its five independent data sets spanning approximately 6 months. Remarkably, this is a very close TDE since the galaxy is in the Virgo Cluster.

The power of in-band spectral fitting paid off for this strong source. Because we had 3 total intensity data points at L-band, 2 total intensity points at C-band, as well as in-band spectral indices for each one of these data points, it was possible to fit a spectrum to the AGN. As shown in Figure 1a, the AGN is a GHz-peaked source (GPS) which is an order of magnitude closer than any GPS source seen before. The spectral peak both declines with time as well as shifts to lower frequencies.

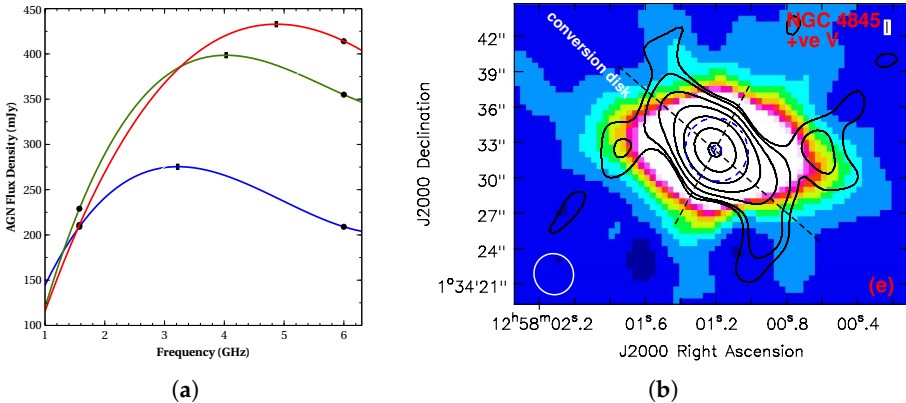

(a)  (b)

**Figure 1.** The galaxy, NGC 4845. (**a**) spectrum of the AGN in NGC 4845 from three different time steps, the earliest in red (T1), T1 + 56 days in green, and T1 + 196 days in blue, from [14]. (**b**) Contours of Stokes V over total intensity image in colour at L-band, from [16]. The beam is shown at the lower left.

The low frequency turnover has been explained by synchrotron self-absorption and the behaviour of the emission modelled by adiabatic expansion of conical or jet-like outflow [14]. Our model further predicts the rate of decline of the radio emission with time which is $S_\nu \propto t^{-5/3}$ for cone-like geometry (Equation (22) of [14]). Follow-up observations 3 and 4 years later show a good fit to the predicted $t^{-5/3}$ decline ([17], and additional as yet unpublished data).

The fact that such a nearby galaxy harbours a TDE-related outburst means that such systems can be monitored with unprecedented resolution. The relationship between classical AGNs and TDEs

may simply be a matter of how much mass is involved in the infall and how sporadically the related outburst occurs. A second TDE has now also been identified in NGC 660 in the X-ray.

### 2.1.2. Circular Polarization

Circular Polarization (CP) is a difficult quantity to measure in extragalactic sources since it is typically very weak ($V/I \approx 0.2\%$) and is technically challenging. Nevertheless, the TDE source, NGC 4845, shows a very strong CP signal of $\approx 2\%$ at L-band near the turnover to optical thickness in the spectrum [14]. This unusual result prompted us to search through all of the galaxies at high resolution (B configuration L-band), resulting in three more detections (NGC 4388, NGC 660, NGC 3628) and a marginal detection in NGC 3079 [16]. In all cases, the detection was right at the core of the galaxy. Weaker signals (less than approx. 0.2%) could potentially be present in the sample but could not have been detected with our technical set-up. The explanation that is most consistent with the data is *Faraday conversion* which converts linear to circular polarization along a line of sight (e.g., [18]). The facts that no linear polarization is observed, that the Stokes V spectrum is very steep ($\alpha_V \approx -3$), and that CP is observed close to the optically thick limit, are in agreement with this interpretation. We provide practical expressions for the Stokes V flux density for a variety of physical conditions in galaxy cores ([16], Appendix A). We also note that the two galaxies with the strongest 2% signals are also those which have the most luminous X-ray cores.

Unlike previous measurements (with the exception of the Galactic Center), CHANG-ES has now revealed *nearby* galaxies with a CP signal, again providing clearer laboratories than before for studying such phenomena. Such a study is only in its infancy but has the potential to reveal activity deep into the core where linear polarization is undetectable. Never-before-seen-related phenomena have now been observed, for example, what we have called the 'conversion disk' in NGC 4845 as seen in Figure 1b. Here, Stokes V contours are shown over the total intensity emission at L-band, revealing a disk that is tilted with respect to the total intensity emission. Presumably, the conversion disk has something to do with the shape of a linearly polarized signal prior to conversion.

### 2.1.3. Hidden AGN-Related Activity Revealed

The polarization capability of CHANG-ES has revealed previously hidden AGN activity which had either been masked or was not as clear in total intensity. A good example is NGC 2992, whose bipolar radio structure is only revealed in polarization [19]. NGC 3079 is another galaxy whose radio lobes are more clearly seen in polarization.

Hidden AGN-related activity can also sometimes be revealed via the spectral index maps. An example is NGC 3628 which shows bipolar radio emission perpendicular to the major axis via flatter spectral indices in comparison to the surrounding emission [5].

These results show the importance of both polarization and spectral index mapping in disentangling AGN-related emission from emission related to other processes.

### 2.1.4. Frequency of AGNs

For the CHANG-ES galaxies, like other spirals, it has historically been a challenge to understand whether outflows might be due to stellar winds and supernovae (SNe) (SF-related activity), or whether the outflow is due to AGNs. This is because there could very well be both types of activity in galaxy nuclei. Theoretically, we expect supermassive black holes to be present in all galaxies (e.g., [20]), although this does not necessarily imply AGN activity. Additionally, low-luminosity AGNs (LLAGNs), when present, can be 'buried' in other types of emission.

Since the CHANG-ES galaxies are edge-on, obscuration can be severe at other wavelengths so adopting radio criteria for AGNs is useful. We use (1) the presence of point-like cores, (2) a luminosity that is too high for a collection of SNe, (3) flat or positive spectral indices at the centre, (4) the presence of bipolar or lobe-like structures, (5) time variability, and (6) CP. Using these criteria where possible, we find an occurrence of 55% of the sample [5]. This is a lower limit because weak LLAGNs could

still be present in some cases and missed by our criteria. Higher resolution observations in the future would help to further distinguish AGNs from central starbursts.

## 2.2. Edge-on Disks and Thermal/Nonthermal Separation

Since we would like to understand the propagation of CREs as a function of position in our galaxies, we need the non-thermal spectral index, $\alpha_{NT}$, as well as the spatially resolved thermal fraction in the CHANG-ES galaxies. To assist with this, we have obtained H$\alpha$ maps of most of our galaxies, with the remainder taken from previously published data. Since both thermal radio emission and H$\alpha$ emission are proportional to the emission measure, EM, the H$\alpha$ maps, corrected for dust, can be used to predict the thermal fraction in the radio. The non-thermal emission can then be found as a function of position in the galaxy.

This issue has been addressed by others in the past (e.g., [21,22]), but not specifically for edge-on galaxies for which dust extinction can be severe. The first indication of this comes from [23] in a plot of L-band luminosity against SFR for the CHANG-ES galaxies, with the SFR determined from the 22 μm luminosity. This plot is essentially the mid-IR radio continuum relation for our galaxies and shows a normalization that differs from face-on systems. That is, for a given radio continuum luminosity, the SFR is *low* by a factor of two to three, suggesting that the 22 μm flux may itself be suffering from some extinction. Thus, standard corrections for dust need to be modified for edge-on or very dusty galaxies.

A variety of tests [24] have shown that, for resolutions of typically 10 arcsec ($\approx$ 1 kpc at 20 Mpc),

$$L\left(\mathrm{H}\alpha_{corr}\right) \,=\, L\left(\mathrm{H}\alpha_{obs}\right) \,+\, 0.042\nu L_{\nu}\left(24\mu\mathrm{m}\right) \tag{1}$$

where $L$ is the luminosity, *corr* and *obs* refer to corrected and observed, respectively, and the 24 μm luminosity (approximately the same as at 22 μm) corrects for dust. The constant is about 1.4 to 2× larger than previous authors and takes the higher extinction into account.

It should be pointed out that in the highest resolution images (3 arcsec, or $\approx$ 300 pc at 20 Mpc), in which HII region complexes in disks start to be resolved, the thermal fraction can be higher than Equation (1) predicts when applied to those discrete HII regions [13].

## 2.3. Vertical Scale Heights, Winds, and Some Unexpected Results

An important goal of the CHANG-ES survey is to measure vertical scale heights in our galaxies, an improvement over simply measuring the radio halo extent to some contour level since scale heights will not be strongly dependent on the sensitivity of the image. Such information (e.g., the ratio of scale heights at two frequencies) can tell us something about the way in which cosmic rays propagate away from the disk, e.g., via diffusion or winds, as outlined via the SPINNAKER code of [25,26]. Careful measurements are required, ensuring that only the highest inclination galaxies and galaxies for which there are no missing large spatial scales are included in the sample. We have therefore measured scale heights in 13 galaxies [27] finding sample averages of 1.4 $\pm$ 0.7 kpc at L-band and 1.1 $\pm$ 0.3 kpc at C-band.

An unexpected result is that, when scale heights are correlated with a variety of parameters (e.g., SFR, B-field strength, etc.), the strongest correlation is actually with galaxy *size*, i.e., larger galaxies have larger radio halos (see Figure 2a). Further analysis suggests a good correlation with global SFR [13]. The physical reason for this correlation and its relation to star formation is discussed in Krause (2019, this volume).

If we remove the dependence on size by forming a *normalized* scale height scale, we now see an *inverse* dependence of the normalized scale height on underlying mass surface density (Figure 2b). That is, as the underlying surface mass increases, the normalized scale heights decrease. This is the first evidence for gravitational deceleration in vertical outflows. Subsequent observations (e.g., NGC 4666 [28]) agree with this result.

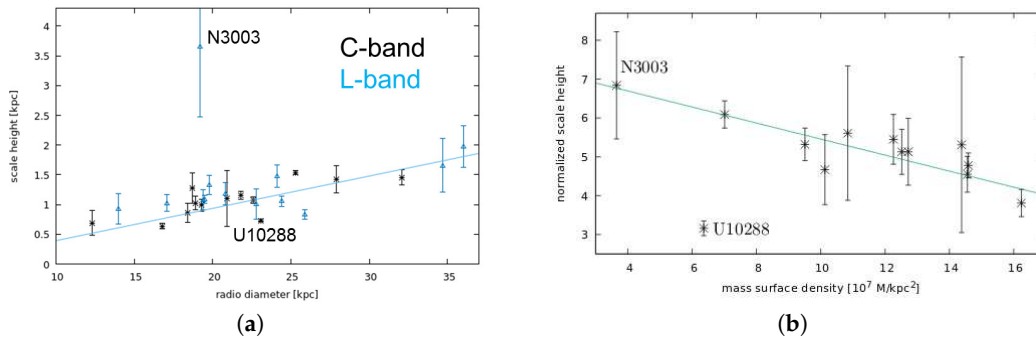

**Figure 2.** Galaxy-scale height correlations from [27]. (**a**) Scale heights increase with the radio diameter of the galaxy. (**b**) Normalized scale heights *decrease* with the mass surface density of the underlying disk.

Once the normalized scale heights are considered, SPINNAKER results show that outflow velocities are consistent with galactic winds with speeds exceeding the escape velocity of the galaxy. Again, subsequent observations [28–31] agree with this.

A beautiful illustration of global halos can also be seen in Figure 3 in which we show the 'median edge-on galaxy' in total intensity modified from [3]. Thirty galaxies have been scaled to the angular size of the largest galaxy, the median formed, and then superimposed on an optical scaled image of the galaxy, NGC 5775.

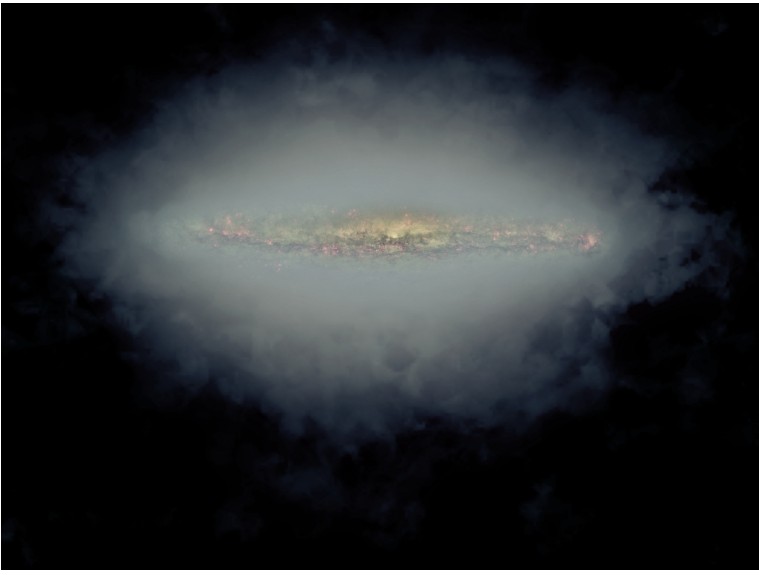

**Figure 3.** The median edge-on spiral galaxy (in blue-grey) in L-band made from stacking 30 of the CHANG-ES galaxies all scaled to the same angular size. The median image is superimposed on an optical Hubble Space Telescope image of NGC 5775. Modified from [3] by Jayanne English (NRAO/AUI/NSF; NASA/STScI).

*2.4. Magnetic Fields and Their Orientation—Reversing Fields*

One of the most remarkable results from the CHANG-ES survey is the discovery of *reversing magnetic fields* in our galaxy disks and halos. Such a careful and detailed look at halo magnetic fields has never before been carried out and is assisted by the application of the Rotation Measure Synthesis (RMS) technique [32]. A number of CHANG-ES galaxies have now had their magnetic fields probed using this technique (e.g., Section 2.5 below).

The most stunning result can be seen in NGC 4631 (Figure 4) which has a well-known strong halo. Here, we see systematic reversals (positive/negative) in the RM (implying systematic reversals

in the magnetic field direction) in the halo of the galaxy on kpc scales. This is the first time that such a regular pattern has been seen in any galaxy halo. Follow-up modeling of NGC 4631 by [33] and earlier papers [34,35] have shown that such fields follow from scale-invariant or self-similar solutions of the classical dynamo equations. The amplitude of these fields is constant on cones in the halo and disc but the field lines are not confined to these cones. Rather, the field lines have quite general poloidal components that frequently loop over the spiral arm projections for the asymmetric modes. The axially symmetric mode also has spiral and poloidal projections. This structure remains a prediction to be tested. However, such field geometry is compatible with the existing observations in CHANG-ES galaxies.

Sign reversals in edge-on disks have now also been observed [28] as well as in other CHANG-ES galaxies (not yet published).

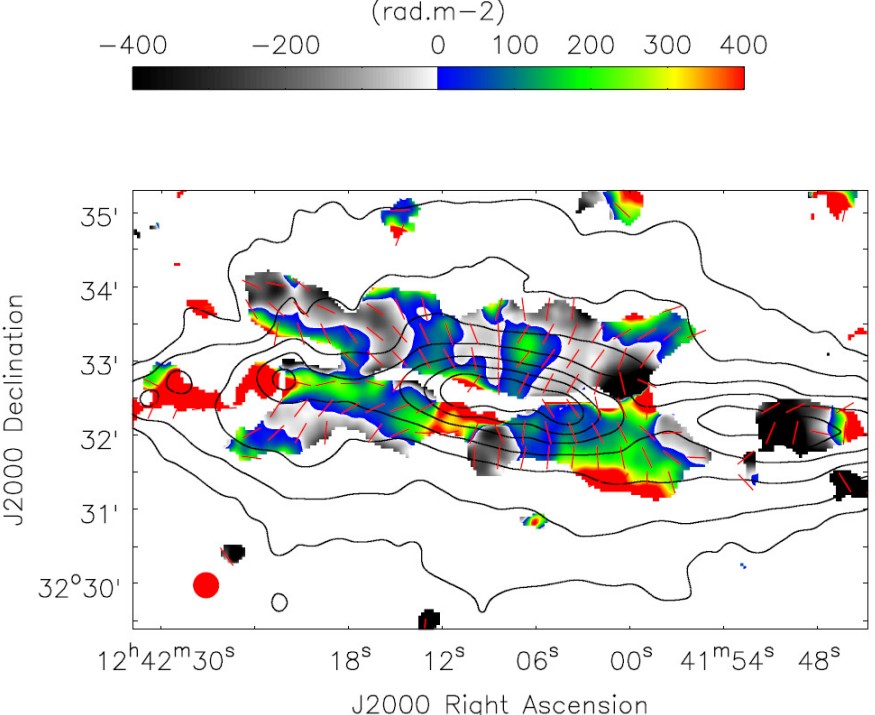

**Figure 4.** Total intensity emission contours over a colour image of RMs in NGC 4631 from [36]. Linear polarization vectors corrected for Faraday rotation are shown in red. Notice the east/west positive/negative RM reversals in the halo to the north of the disk. The colour scale at top specifies the RM values. The beam is shown as a red dot at the lower left.

### 2.5. Individual Galaxies

In this section, we focus on some individual galaxies that have received attention from the CHANG-ES consortium so far. They were studied in some detail for a variety of reasons, some of which are related to the PhD programs of our student members.

#### 2.5.1. UGC 10288

UGC 10288 (Figure 5) was studied in some detail early in the project [37]. For this galaxy, we combined all five total intensity data sets to obtain a single image at a frequency intermediate between L-band and C-band (4.1 GHz). The fact that in-band spectral index fitting occurred at every position in the galaxy (Section 1.2) allowed such a combination to be carried out.

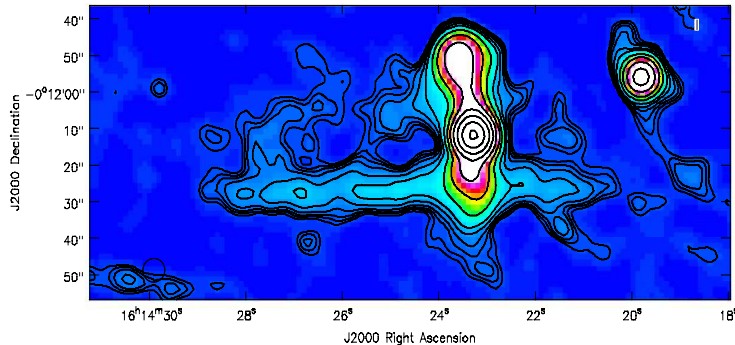

**Figure 5.** UGC 10288 after combining all data sets, resulting in a central frequency between L-band and C-band (4.1 GHz). The strong vertical feature near the centre is a background double-lobed source. The beam (not shown) is 6.5 arcsec in size.

Surprisingly, most of the emission from the location of this galaxy is due to a bright background double-lobed radio source ('CHANG-ES A') whose central AGN is situated just above the disk of UGC 10288 and whose southern radio lobe shines right through the disk. The radio emission from UGC 10288 itself would have fallen below our flux density limit for CHANG-ES, had the background galaxy flux density been subtracted off. This result warns that care is required in interpreting discrete features that appear to be in the galaxy but may in fact be background sources, especially for such sensitive observations (see also [5]).

In spite of the corresponding downwards revision of the SFR for UGC 10288 ($\approx$0.4 $M_\odot$/yr), vertical discrete radio continuum features can still be seen extending out of the disk of this galaxy. A preliminary RM analysis of the light from the background source as it passes through the disk and halo region of UGC 10288 gave hints of RM reversals, though not with the stunning regularity shown for NGC 4631 in Figure 4.

### 2.5.2. NGC 4666

The CHANG-ES radio continuum data of the superwind and starburst galaxy NGC 4666 (Figure 6 and [28]) show a boxy radio halo in total intensity as well as in the X-ray. The wind is likely driven by supernovae across almost the entire disk. For the first time, the central point source was observed at radio wavelengths. In our X-ray data, clear AGN fingerprints were detected, which supports previous findings from X-ray observations that NGC 4666 harbors an AGN.

The high resolution combined C-band data of NGC 4666 (Figure 6) show remarkable filamentary vertical structures reaching into the halo-like fingers. A scale height analysis reveals a thick disk with a mean scale height of 1.57 $\pm$ 0.21 kpc (C-band) and 2.16 $\pm$ 0.36 kpc (L-band). The CR transport was modelled with the SPINNAKER code and found to have an advective wind of 310 km s$^{-1}$. This is different from the CHANG-ES galaxy NGC 4013 [38], where diffusion seems to be the dominant CR transport process.

NGC 4666 is characterized by an extended X-shaped structure with magnetic field vectors reaching far into the halo. The mean disk magnetic field strength is 12.3 µG, which is in good agreement with the field strength of other spiral galaxies [39]. The analysis of the C-band RM-synthesis map along the major axis of NGC 4666 shows indication of one disk magnetic field reversal at a radius of 30″ (4 kpc). The field orientation is likely axisymmetric and points inwards and then outwards with increasing radial distance from the centre (please see [28] for more details on the analysis). This is the first time that an indication of a radial field reversal within the disk is observed in an external galaxy. As we saw for the magnetic field reversals in the halo of NGC 4631 (Section 2.4), these findings are consistent with mean-field dynamo theory (e.g., [40]).

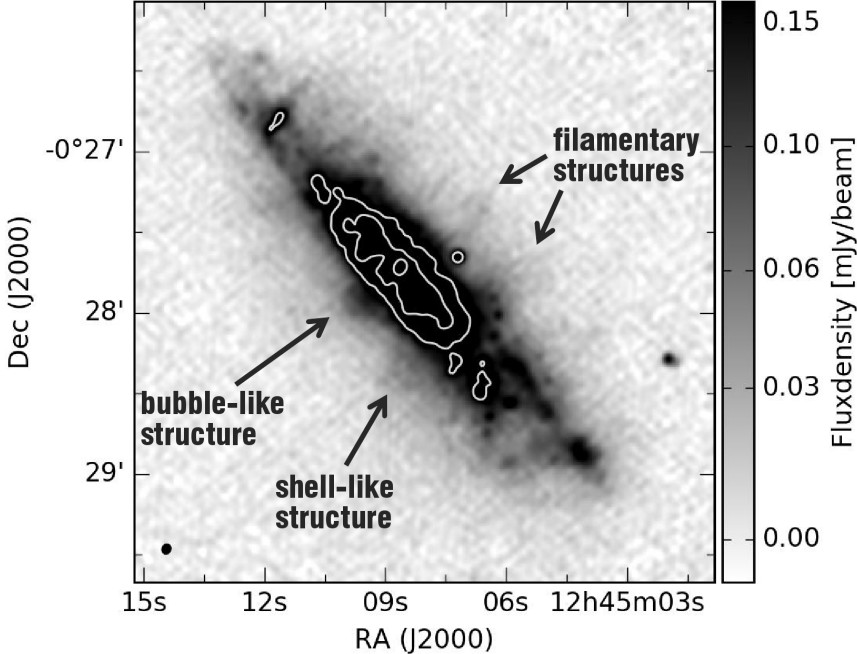

**Figure 6.** C-band image of NGC 4666 in gray scale and contours, the latter at 0.22, 0.44, and 0.88 mJy beam$^{-1}$ with a beam (shown at the lower left) of $3.0'' \times 3.5''$. The central contour depicts the AGN in this galaxy and other features are labelled. From [28].

### 2.5.3. NGC 4388

In the case of this Virgo galaxy, the sensitivity provided by the use of the RM-synthesis technique on our C band-C configuration data [41] was key to revealing a large number of new faint features in the halo (Figure 7). With an exceptionally low rms noise of 2.3 µJy/beam and a resolution of 5.33 arcsec in polarized intensity, we are also able to resolve some interesting features of the disk never seen before.

This is a Seyfert 2 galaxy with many processes occurring at the same time. There are several outflow events detected at different wavelengths (i.e., Hα, X-ray, HI) whose kinematics and morphology are strongly modified by the strong ram-pressure action of the intracluster medium (ICM). The brightest emission in radio polarization in our VLA map is detected at the northern part of the nuclear outflow, where the magnetic field vectors are oriented along the outflow structure. Our data also show for the first time a well defined southern nuclear outflow where the magnetic field vectors are distributed in an S-shaped structure, which is an indication of a connection between the bright radio spots detected above the disk and the nucleus of the galaxy. The halo also shows two polarization filaments to the north and south of the nuclear outflow that extend up to ∼ 5 kpc in the case of the northern filament. These structures seem to also be connected to the nuclear activity due to its symmetry with respect to the centre. It is still unclear whether their origin is AGN- or star formation-related.

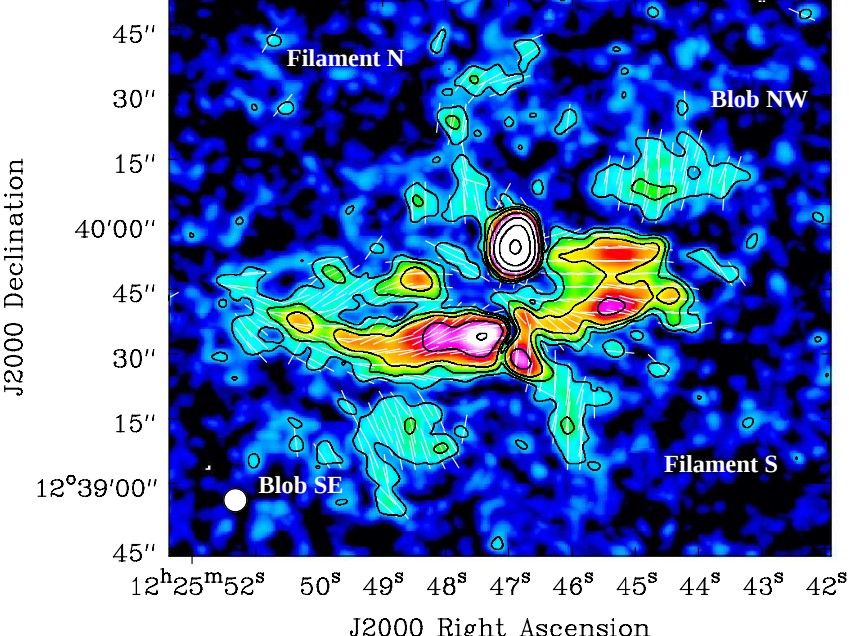

**Figure 7.** Linearly polarized intensity at 6 GHz in contours plus magnetic vectors corrected for Faraday rotation, with a resolution of 5.33 × 5.33 arcsec and an rms noise of 2.3 μJy beam$^{-1}$. The contour levels are (3, 5, 8, 16, 32, 64, 128) × 2.3 mJy beam$^{-1}$. The beam is shown as a white dot at the lower left.

NGC 4388 shows two extended blobs of polarized signal towards the southeast and the northwest parts of the halo and their magnetic field vectors are oriented perpendicular to the disk of the galaxy. By computing the half-life time of the CREs in these regions, we can estimate how far into the halo these particles can reach before losing all their energy. We found that in order to reach heights between 2.9 and ∼ 3.7 kpc, they would need to travel at a speed of 270 km s$^{-1}$, which is typical for a galactic wind.

### 2.5.4. NGC 3556 and LOFAR Data

Total power results from CHANG-ES observations of NGC 3556 were combined with results from 144 MHz observations carried out using the Low Frequency Array (LOFAR) [29]. These data were taken as part of the LOFAR Two-metre Sky Survey (LoTSS) [42]. The combined results were used to calculate the spectral index between L-band and 144 MHz and the magnetic field strength in the galaxy using the equipartition approach by [43]. This showed that the disk of the galaxy has an average magnetic field strength of 9 μG, decreasing to around 5 μG towards the halo. The magnetic field strength map is presented in Figure 8.

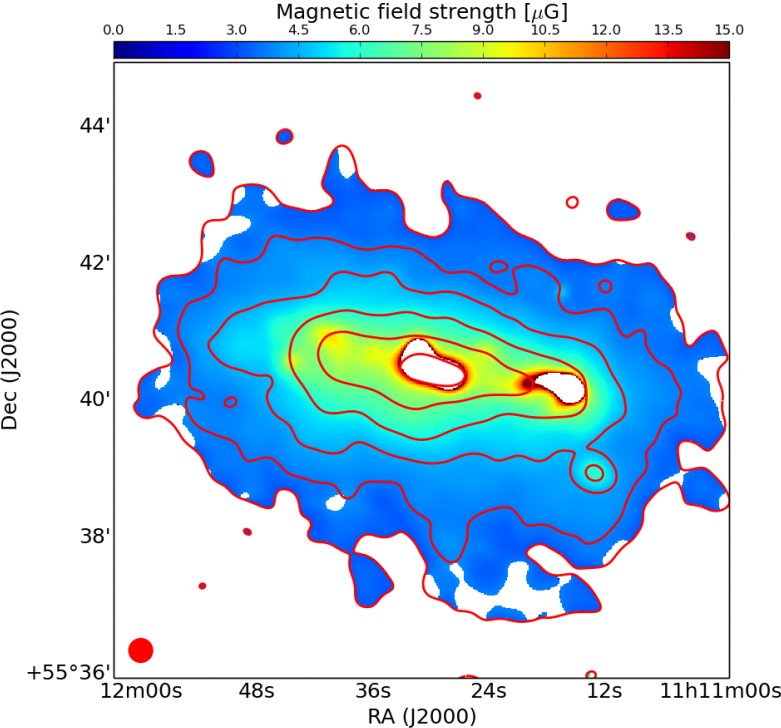

**Figure 8.** Magnetic field strength map of NGC 3556 derived through the equipartition assumption using the spectral index between *L* band and 144 MHz and synchrotron flux density at 144 MHz. Red contours show the surface brightness distribution of the 144 MHz map. The beam is shown as a red dot at the lower left.

The total power values and the magnetic field strength were used to model the cosmic ray electron propagation using SPINNAKER [25] and its interactive wrapper SPINTERACTIVE [3]. Multiple models of diffusive and advective propagation were tested. The best fitting result was an advective transport of the CREs with a linearly increasing wind speed.

2.5.5. NGC 891 and NGC 4565

The two nearby spiral galaxies, NGC 891 and NGC 4565, differ largely in their detectable halo extent and their star formation rates. The question is how these differences are related to the (advective and/or diffusive) CRE transport in the disk and halo. In C-band, the total power emission of both galaxies has been corrected for missing zero spacing flux with the single-dish 100-m Effelsberg telescope. After subtracting off the thermal emission, maps of the total magnetic field strength and the nonthermal spectral indices as well as vertical nonthermal scale heights at different radii could be determined [31].

These quantities were used to model the CRE transport by SPINNAKER. Schmidt et al. [31] found that the CRE transport in the halo of NGC 891 is probably dominated by advection with an accelerated galactic wind reaching escape velocity at a height of 9 to 17 kpc. This wind is likely to coexist with diffusion-dominated regions. For NGC 4565, however, the halo is diffusion-dominated up to a height of at least 1 kpc with a diffusion coefficient being only weakly energy dependent.

The high-resolution total intensity radio map of NGC 4565 in C-band has a linear resolution of about 200 pc that allows the possibility of separating both sides of the large radio ring (which could

---

3    https://github.com/vheesen/Spinnaker

also represent tightly wound inner spiral arms) in the disk, as shown in Figure 9. This ring-like feature must be very thin with a vertical thickness of only $260 \pm 100$ pc and horizontal width of $2.2 \pm 2.2$ kpc.

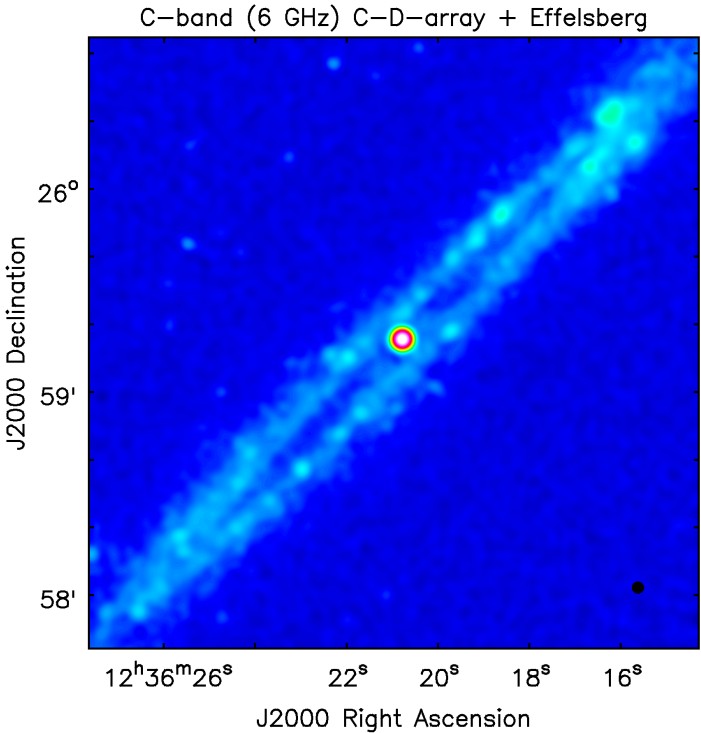

**Figure 9.** The galaxy, NGC 4565, in total intensity at C-band and resolution of 200 pc, zooming in on the large, thin, ring or inner spiral arms. From [31]. Notice the point-like core at the centre. The beam is shown as a small black dot at the lower right.

## 3. Conclusions

The CHANG-ES sample of 35 edge-on galaxies has been observed in wide bands centred at 1.6 and 6.0 GHz over a variety of VLA array configurations. This survey, which includes spatially resolved polarization and spectral index information, is providing new insight into the halo and disk-halo activity in spiral galaxies.

With thermal/non-thermal separation of the emission, measurements of scale-heights, and the sophisticated modeling of CRE outflows, the physical conditions in such regions are being probed in considerably more detail than ever before. The results of such modeling suggest that most of our galaxies are experiencing outflowing winds.

AGNs have been revealed in the CHANG-ES sample in a variety of ways, including in circular polarization. Bipolar features can be seen in polarization or spectral index structures which otherwise have been masked in total intensity.

The polarization data together with a rotation measure synthesis analysis have now revealed structures that have never before been seen in any galaxy. These are regularly reversing magnetic fields on kpc scales both in the disk and halo, likely because of magnetic dynamo action.

We summarize CHANG-ES papers to date in Table 2. CHANG-ES FITS images are available at the data release web site: https://www.queensu.ca/changes.

**Table 2.** List of CHANG-ES papers to date.

| Paper Title | Ref |
| --- | --- |
| CHANG-ES I: Continuum Halos in Nearby Galaxies: An EVLA Survey: Introduction to the Survey | [4] |
| CHANG-ES II: Continuum Halos in Nearby Galaxies: An EVLA Survey: First Results on NGC 4631 | [12] |
| CHANG-ES III: UGC 10288—An Edge-on Galaxy with a Background Double-lobed Radio Source | [37] |
| CHANG-ES IV: Radio continuum emission of 35 edge-on galaxies observed with the Karl G. Jansky Very Large Array in D-configuration — Data Release 1 | [3] |
| CHANG-ES V: Nuclear Outflow in a Virgo Cluster Spiral after a Tidal Disruption Event | [14] |
| CHANG-ES VI: Probing Supernova energy deposition in spiral galaxies through Multiwavelength relationships | [23] |
| CHANG-ES VII: Magnetic Outflows from the Virgo Cluster Galaxy NGC 4388 | [41] |
| CHANG-ES VIII: Uncovering Hidden AGN activity in radio polarization | [19] |
| CHANG-ES IX: Radio scale heights and scale lengths of a consistent sample of 13 spiral galaxies seen edge-on and their correlations | [27] |
| CHANG-ES X: Spatially Resolved Separation of Thermal Contribution from Radio Continuum Emission in Edge-on Galaxies | [24] |
| CHANG-ES XI: Circular polarization in the cores of nearby galaxies | [16] |
| CHANG-ES XII: A LOFAR and VLA View of the Edge-on Star Forming Galaxy, NGC 3556 | [29] |
| CHANG-ES XIII: Transport processes and the Magnetic Fields of NGC 4666 — Indication of a Reversing Disk Magnetic Field | [28] |
| CHANG-ES XIV: Cosmic-ray Propagation and Magnetic Field Strengths in the Radio Halo of NGC 4631 | [36] |
| CHANG-ES XV: Large-scale Magnetic Field Reversals in the Radio Halo of NGC 4631 | [30] |
| CHANG-ES XVI: An In-Depth View of the Cosmic-ray Transport in the Edge-on Spiral Galaxies NGC 891 and NGC 4565 | [31] |
| CHANG-ES XVII: H-alpha Imaging of Nearby Edge-on Galaxies, New SFRs, and an Extreme Star Formation Region — Data Release 2 | [13] |
| CHANG-ES XVIII: The CHANG-ES Survey and Selected Results | [44] |
| CHANG-ES XIX: The Galaxy NGC 4013 — A diffusion-dominated radio halo with plane-parallel disk and vertical halo magnetic fields | [38] |
| CHANG-ES XX: High Resolution Radio Continuum Images of Edge-on Galaxies and their AGNs — Data Release 3 | [5] |
| CHANG-ES XXI: Radio continuum emission of 35 edge-on galaxies observed with the Karl G. Jansky Very Large Array in C-configuration — Data Release 4 | [6] |

**Author Contributions:** As this article highlights contributions that are largely published or in press, the CHANG-ES consortium as a whole is to be credited. For authors on this particular paper, each one made contributions to the paper as a whole, providing suggestions for improvements and corrections where needed. In addition, some authors focussed on a section or sections or other specifics of this paper. The following are some examples that are not meant to exclude the particular co-author from other forms of involvement. J.I.: Principal Investigator and primary author; A.D.-S.: NGC 4388; M.K.: scale-heights and NGC 4631; A.M.: NGC 3556; J.L.: galaxy correlations; Y.S.: NGC 4666; J.E.: visualization; R.H.: Magnetic dynamo modelling; R.B.: methodology and validation; T.W.: data reduction and display; R.-J.D.: validation.

**Funding:** For the first author, this research was funded by the Natural Sciences and Engineering Research Council of Canada Discovery grant number 388-528.

**Acknowledgments:** The first author wishes to thank the Natural Sciences and Engineering Research Council of Canada for a Discovery Grant.

**Conflicts of Interest:** The authors declare no conflict of interest.

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
