# Peer review of "CHANG-ES: XVIII—The CHANG-ES Survey and Selected Results"

_galaxies, doi:10.3390/galaxies7010042_

Round 1

Reviewer 1 Report

This manuscript reviews some selected results of the KJ-VLT CHANG-ES survey at 1.58 and 6 GHz. The polarization observations allow to investigate the magnetic field properties of the galaxies, including their nuclei, halos and outflows.

The manuscript examines the CHANG-ES results by topic, which is appropriate to present them.

COMMENTS

I miss a more general introduction. The paper starts with the CHANG-ES survey description and only comments that it has 'an order of magnitude improvement, on average, compared to any similar survey.' It would be more helpful if the authors provide a short description of these similar surveys, their limitations and shortcomings, and then why the CHANG-ES survey is necessary and how it is planed to improve previous research.

At least 3/4 of the introduction looks like methodological and sample selection issues. A description of the VLA array configurations used, at least including resolutions more detailed than in table 1 would be helpful.

A short explanation about the 'multi-scale multi-frequency synthesis' would also be helpful.

Also a short explanation about 'curvature in the spectral index' would be useful, or if it was not used, simply say that S/N was not high enough to justify higher order corrections to the ms-mfsfittings.

I found confusing the statement: 'Release data include total intensity images at two differentuvweightings...' There were not enough explanations about BCD configurations anduvweightings to understand what the authors exactly mean.

Please, include a note to identify the resolution/beam size in figures 1, 4, 5 (I think that the beam profile is superposed to another feature in fig. 5), 7, 8 and 9 (is the tiny small black dotat the bottom-right of this figure the beam size?orit is the central white dot at the center of the ring?, and is the nucleus subtracted?). A numerical value of the resolutionineach image (particularly on thosewerethe beam is not shown) would also be helpful.

Figure 4. Explain the scale above the figure.

Galaxy NGC 4565. The text says 'NGC 4565 in C-band has a linear resolution of about 200pcthat allows the possibility of separating both sides of the large radio ring or spiral arms in the disk as shown in Figure 9.' However, Figure 9 only shows the radio ring, but not the spiral arms. 

Finally, please check for references 31 and 37, either they are not present in the text or I missed them.

Author Response

RESPONSE TO REFEREE 1:

This manuscript reviews some selected results of the KJ-VLT CHANG-ES survey at 1.58 and 6 GHz. The polarization observations allow to investigate the magnetic field properties of the galaxies, including their nuclei, halos and outflows.

The manuscript examines the CHANG-ES results by topic, which is appropriate to present them.

COMMENTS
I miss a more general introduction. The paper starts with the CHANG-ES survey description and only comments that it has 'an order of magnitude improvement, on average, compared to any similar survey.' It would be more helpful if the authors provide a short description of these similar surveys, their limitations and shortcomings, and then why the CHANG-ES survey is necessary and how it is planed to improve previous research.

***********************************
A comparison with previous surveys rms values is now included at the end of the first paragraph. The goals are summarized after the table.
**********************************

At least 3/4 of the introduction looks like methodological and sample selection issues. A description of the VLA array configurations used, at least including resolutions more detailed than in table 1 would be helpful.

*************************************
We have broken up the resolution information into the different data sets and added the values to the table.
************************************

A short explanation about the 'multi-scale multi-frequency synthesis' would also be helpful.

***************************************
Sentences describing the ms-mfs algorithm have been added.  
**************************************

Also a short explanation about 'curvature in the spectral index' would be useful, or if it was not used, simply say that S/N was not high enough to justify higher order corrections to the ms-mfsfittings.

*************************************
Modified text has been included.

***********************************

I found confusing the statement: 'Release data include total intensity images at two differentuvweightings...' There were not enough explanations about BCD configurations anduvweightings to understand what the authors exactly mean.

********************************
The weightings are now specified.
********************************

Please, include a note to identify the resolution/beam size in figures 1, 4, 5 (I think that the beam profile is superposed to another feature in fig. 5), 7, 8 and 9 (is the tiny small black dotat the bottom-right of this figure the beam size?orit is the central white dot at the center of the ring?, and is the nucleus subtracted?). A numerical value of the resolutionineach image (particularly on thosewerethe beam is not shown) would also be helpful.

************************************

Beams in Fig. 1, 4, and 5 are now specified. For Fig. 5, you are probably looking at a strong unresolved source to the upper right which, since unresolved, looks like the beam pattern.  Beams are now specified in Fig. 7, 8 and 9.  Yes the tiny black dot is the beam, now specified, for Fig. 9 and we note the central point-like core in the caption.  In fact, the beams were shown on every figure except Fig. 5 which we now specify numerically.

***********************************

Figure 4. Explain the scale above the figure.

*************************************
Now specified.
************************************

Galaxy NGC 4565. The text says 'NGC 4565 in C-band has a linear resolution of about 200pcthat allows the possibility of separating both sides of the large radio ring or spiral arms in the disk as shown in Figure 9.' However, Figure 9 only shows the radio ring, but not the spiral arms.

*************************************
Wording has been clarified.
*************************************

Finally, please check for references 31 and 37, either they are not present in the text or I missed them.

************************************
Ref. 31 is cited in section 2.5.4, Ref. 37 is cited in Sect. 1.1, par starting "Our galaxy selection..."
***********************************

Reviewer 2 Report

Referee Report:
CHANG-ES XVIII - The CHANG-ES Survey and Selected Results
Irwin et al.

Report:
This paper is a review of the previously published results from the CHANG-ES survey. No new results are presented. As such, there is comparatively little to actually review (since there’s no point digging deeply into results that are already published elsewhere). The review is well written and seems to cover all the major results of the survey. I recommend publication, with only a few small suggestions for things I think would improve the paper slightly.

Major comments:
While completely optional, one thing that I would like to see in this paper would be a table listing all the CHANG-ES papers and their most interesting science result. I think it would help drive home the message of how much science has come out of the survey.

I think the conclusions section could benefit from an additional paragraph describing the future of the survey: what data products are planned to be released, and on what time-scale? What are the logical directions to continue (e.g., more LOFAR comparisons? More observations covering different areas of observational parameter space?)?

Minor comments:
Fig 4: the polarization vectors are de-rotated, right? Please explicitly state so.

L261: does it show a magnetic field reversal, or a reversal in the sign of RM? Granted one implies the other, but I won’t necessarily agree that they will always be co-located depending on the geometry of the line of sight (especially if this is in the disk, which is not clear from the current wording).
L264-265: ‘consistent with dynamo theory’ is a very vague phrase, given how many variations of dynamo theory there are and how many different predictions can be/have been made. I’d like it if a more specific dynamo model could be named.
L286: space between km and s

English comments:
L41: was -> were
L156: types emission -> types of emission

Author Response

RESPONSE TO REFEREE 2
Referee Report:
CHANG-ES XVIII - The CHANG-ES Survey and Selected Results
Irwin et al.

Report:
This paper is a review of the previously published results from the CHANG-ES survey. No new results are presented. As such, there is comparatively little to actually review (since there’s no point digging deeply into results that are already published elsewhere). The review is well written and seems to cover all the major results of the survey. I recommend publication, with only a few small suggestions for things I think would improve the paper slightly.

Major comments:
While completely optional, one thing that I would like to see in this paper would be a table listing all the CHANG-ES papers and their most interesting science result. I think it would help drive home the message of how much science has come out of the survey.

***************************
I think the best way to pursue this is to include a table giving all papers and their titles which we've now done in Table 2.  The most important results are, of course, given in this paper, thoroughly referenced.
****************************

I think the conclusions section could benefit from an additional paragraph describing the future of the survey: what data products are planned to be released, and on what time-scale? What are the logical directions to continue (e.g., more LOFAR comparisons? More observations covering different areas of observational parameter space?)?

****************************
This is actually a matter of discussion amongst our group and it's premature to commit on paper.  I hope you can let this one go.
***************************

Minor comments:
Fig 4: the polarization vectors are de-rotated, right? Please explicitly state so.

**************************
done
**************************

L261: does it show a magnetic field reversal, or a reversal in the sign of RM? Granted one implies the other, but I won’t necessarily agree that they will always be co-located depending on the geometry of the line of sight (especially if this is in the disk, which is not clear from the current wording).
L264-265: ‘consistent with dynamo theory’ is a very vague phrase, given how many variations of dynamo theory there are and how many different predictions can be/have been made. I’d like it if a more specific dynamo model could be named.

***************************
See Sect. 2.5.2 for wording adjustments and a reference to the relevant model.
***************************

L286: space between km and s

************************
fixed
**************************

English comments:
L41: was -> were
L156: types emission -> types of emission
*****************************
both fixed
*******************************

Reviewer 3 Report

The paper:CHANG-ES:XVIII - The CHANG-ES Survey and Selected Results

gives an overview of the project.

I have only two minor points to rise:

Page 5 line 161: please change ... an occurrence of 55%... with ...an occurrence of 55% of the total sample...

Page 8 line 230: It will be very usefull a brief introduction on the reason that says how the individual cases described have been chosen.

Author Response

RESPONSE TO REFEREE 3

I have only two minor points to rise:

Page 5 line 161: please change ... an occurrence of 55%... with ...an occurrence of 55% of the total sample...

*******************************
This has been fixed
********************************

Page 8 line 230: It will be very usefull a brief introduction on the reason that says how the individual cases described have been chosen.

******************************
An explanatory note has been added
********************************